

# FASDD: An Open-access 100,000-level Flame and Smoke Detection Dataset for Deep Learning in Fire Detection

Ming Wang[1], Liangcun Jiang[1, 2]*, Peng Yue[1, 3, 4, 5]*, Dayu Yu[1], Tianyu Tuo[1]

[1] School of Remote Sensing and Information Engineering, Wuhan University, Wuhan, Hubei, 430079, China

[2] School of Resources and Environmental Engineering, Wuhan University of Technology, Wuhan, Hubei, 430070, China

[3] Hubei LuoJia Laboratory, Wuhan, Hubei, 430079, China

[4] Collaborative Innovation Center of Geospatial Technology, Wuhan, Hubei, 430079, China

[5] Hubei Province Engineering Center for Intelligent Geoprocessing (HPECIG), Wuhan University, Wuhan, Hubei, 430079, China

*Correspondence to*: Liangcun Jiang (jiangliangcun@whut.edu.cn); Peng Yue(pyue@whu.edu.cn)

**Abstract.** With the advancement of computer vision, artificial intelligence, and remote sensing technologies, deep learning algorithms are increasingly used in terrestrial, airborne, and spaceborne-based fire detection systems. The performance and generalization of these data-driven fire detection algorithms, however, are restricted by the limited number and source of fire

detection datasets. A large-scale fire detection benchmark dataset covering complex and varied fire scenarios is urgently needed. This work constructs a 100,000-level Flame and Smoke Detection Dataset (FASDD) based on multi-source heterogeneous flame and smoke images. It holds rich variations in image size, resolution, illumination (day and night), scenario (indoor and outdoor), image range (far and near), viewing angle (top view and side view), platform (surveillance cameras, drones, and satellites), and data source (Internet, social media, and open-access fire datasets). To the best of our knowledge,

FASDD is currently the most versatile and comprehensive dataset for fire detection. It provides a challenging benchmark to drive the continuous evolution of fire detection models. Additionally, we formulate a unified workflow for preprocessing, annotation, and quality control of fire samples. Out-of-the-box annotations are published in four different formats for training deep learning models. Extensive performance evaluations based on classical methods show that most of the object detection models trained on FASDD can achieve satisfactory fire detection results, and especially YOLOv5x achieves nearly 80%

mAP@0.5 accuracy on heterogeneous images spanning two domains of computer vision and remote sensing. And the application in wildfire location demonstrates that deep learning models trained on our dataset can be used in recognizing and monitoring forest fires. Deep learning models trained with FASDD can be simultaneously deployed on satellites, drones, and ground sensors, thus realizing collaborative fire observation and detection in a space-air-ground integrated network environment. The dataset is available from the Science Data Bank website at https://doi.org/10.57760/sciencedb.j00104.00103

(Wang et al., 2022a).



## 1 Introduction

Fire is one of the most severe disasters that threaten human safety and Earth ecology (Gaur et al., 2020; Gibson et al., 2020; Shamsoshoara et al., 2021). Extreme forest fire accidents can cause severe economic losses and devastating ecological damage, and even endanger human life (Chowdary et al., 2018; Gaur et al., 2020). A recent review article by Jones et al. (2022) shows that the length and frequency of global fire weather have increased in recent decades. The seven-month-long Australian bushfire emergency in 2019, as a representative of extreme fire disasters, leaves a deep imprint on the Earth. Fire detection is a very crucial task in the pre-suppression process. Numerous terrestrial, airborne, and spaceborne-based systems equipped with visible, infrared (IR), or multispectral sensors have been developed for fire detection. Visible bands are used to detect smoke, near-infrared and shortwave infrared bands are used to detect reflected light, and middle IR and thermal IR bands are used to measure thermal radiation. Fire detection that relies on a single sensor/platform usually has low sensitivity or a high false alarm rate. Thus, the data fusion technique and the idea of the collaborative observation network have been proposed for smoke and flame detection. The former one focuses on the combined use of data from different types of sensors (e.g. optical and IR sensors), while the latter one attends to building large networks of low-cost optical cameras. For the latter solution, many networks of optical cameras have already been deployed in recent years (Govil et al., 2020), which usually integrate computer vision (CV) and machine learning algorithms for fire detection and localization (Barmpoutis et al., 2020). However, in complex real-world environments, flame and smoke have multiple characteristics of flickering, growth, disorder, various colors, and variable intensity (Muhammad et al., 2018). Flame is also easily confused with many objects such as lights, sun, and maple leaves, and smoke is easily confused with clouds, waterfalls, and hair (Ko et al., 2012; Foggia et al. 2015; Chino et al., 2015). When coupled with the low signal-to-noise ratio scene, it brings additional difficulties to vision-based fire detection methods (Muhammad et al., 2018).

Vision-based fire detection methods mainly include static feature-based, dynamic feature-based, traditional machine learning-based, and neural network-based methods. Static feature-based methods usually implement fire discrimination based on representative features of flame and smoke such as color features (Foggia et al. 2015; Calderara et al. 2008). These static feature-based methods have lower computational costs, yet they also bring lower reliability and higher false alarm rates (Muhammad et al., 2018). Dynamic feature-based methods analyze flame and smoke videos based on flicker (Töreyin et al. 2005), motion, and dynamic texture or the evolution of spatiotemporal information (Dimitropoulos et al. 2015). These methods employ the irregularity and growth properties of flame and smoke, which can improve the detection accuracy to some extent, yet it requires high computational costs. Traditional machine learning-based methods perform fire detection with classical classifiers such as decision tree, support vector machines, and random forest, which are usually trained based on handcrafted features (Chi et al. 2017; Wang et al. 2017). However, these methods face the feature selection bias problem and usually have a high operational complexity and time cost. In this context, neural network-based fire detection methods are emerging. Dua et al. (2020) detect fires based on deep convolutional neural networks (DCNN) and the Transfer Learning approach, which



outperforms traditional machine learning models. Cheng et al. (2019) use the generative adversarial network (GAN) to predict
the changing trend of smoke and improve the smoke segmentation accuracy based on Deeplabv3+ and DenseCRF.

Neural network-based methods have gradually developed into mainstream fire detection methods, which can generally achieve satisfactory detection accuracy. Considering that visual features of flame and smoke have significant differences in different scenes, robust deep learning models usually require large-scale, high-quality training samples to drive (Torralba et al., 2011). Existing open-access datasets for fire detection are oriented to specific sensors (spaceborne, airborne, or terrestrial-
based sensors), specific tasks (such as scene classification, object detection, and semantic segmentation), or specific scenarios (such as indoor fires and wildfires). Those datasets have some limitations such as the small number of samples, fixed image size or resolution, single data source, poor task compatibility, and similar scenes. Meanwhile, the development of space-air-ground integrated networks has shown its potential use in the fire detection domain. It would be helpful for early fire detection if the same deep learning model is deployed on such an integrated network. Thus, there is an urgent need to establish a dataset
with a large amount of heterogeneous flame and smoke samples from multiple sources. Such a dataset shall be produced using a unified specification and managed following FAIR (findability, accessibility, interoperability, and reusability) principles (Geetha et al., 2021).

In this paper, a large-scale heterogeneous Flame and Smoke Detection Dataset (FASDD) is provided, which includes fire data from multiple sources and various scenarios. To overcome the limitations of existing datasets, we collected and carefully
selected a large number of fire images captured by spaceborne, airborne, and terrestrial sensors, which can provide data support for training robust fire detection models and space-air-ground integrated fire detection. The main contributions of this paper are briefly summarized as follows: (1) A 100,000-level flame and smoke detection dataset is constructed. To the best of our knowledge, it is the largest open-access fire dataset with the most complexity in fire scenes, the highest heterogeneity in image feature distribution, and the most significant difference in image size and shape. It can support object detection and
classification tasks in different fire scenes captured by various optical sensors. (2) The dataset is generated according to a unified data model. Moreover, the annotation files are provided in four common dataset formats for FASDD to support different deep learning models. (3) Extensive performance comparison and evaluation based on representative object detection methods are performed on FASDD to provide a valuable reference for using our dataset.

## 2 Related work

### 2.1 Existing fire detection datasets

There are many works on datasets for fire detection. Jakovcevic et al. (2010) first propose a wildfire smoke dataset for the smoke segmentation task, which focuses on smoke in the wild. For the smoke classification task, Yuan (2011) provides a dataset that includes real-time smoke, synthetic smoke, non-smoke images, and videos. Chino et al. (2015) present a flame and smoke dataset that includes 240 training samples and 226 test samples. However, these datasets have a small sample size and
are only applicable to simple classification tasks without accurate bounding boxes or mask labels. There are also some datasets



produced based on videos. Ko et al. (2012) publish a wildfire smoke video dataset. Foggia et al. (2015) provide an influential flame and smoke video dataset containing videos captured indoors and outdoors, during day and night, and at different distances. Zhang et al. (2018) introduce a wildfire smoke video dataset from watchtowers and UAVs (unmanned aerial vehicles). Shamsoshoara et al. (2021) describe a dataset for forest fire detection containing flame and smoke videos and aerial
images captured by infrared cameras. Yet, there are many similar frames in these video datasets, and their heterogeneity and generalizability are insufficient. Sharma et al. (2017) propose a flame image dataset containing flame images with different lighting intensities and scenes. Dunnings et al. (2018) from Durham University publish a flame dataset for the segmentation task, whose image size is set uniformly to 224 × 224 pixels. The image size in these datasets is relatively fixed and small, which cannot fully represent real-world fire scenarios. Geng et al. (2020) provide a large dataset of flame and smoke for object
detection tasks, but most of these images are unlabeled. These available flame and smoke datasets have some limitations in terms of quantity, resolution, and scene (Geetha et al., 2021), but they provide valuable references for developing a large-scale cross-domain fire detection dataset with different scenes and rich characteristics. In addition, most of the current fire detection methods in object detection tasks use UAV (Unmanned Aerial Vehicle) data rather than satellite data (Zhan et al., 2021; Esfahlani 2019), and most of the satellite data are used for fire detection in classification tasks (Shanmuga priya and Vani, 
2019) or semantic segmentation tasks (Rashkovetsky et al., 2021; Wang et al., 2022b). Therefore, fire datasets from spaceborne sensors for object detection tasks are currently scarce, yet our dataset can provide some contribution to this gap.

### 2.2 Annotation tools

An appropriate data annotation tool is beneficial to optimize the data annotation process and improve the data annotation efficiency (Geetha et al., 2021). Image annotation tools for object detection can be divided into offline tools and online tools
(Pande et al., 2022). Offline tools have high autonomy and controllability, which can ensure that data collection, cleaning, labeling, and training are implemented in a local network-free environment. LabelImg (Tzutalin, 2015) is widely used as image annotation software for object detection. It supports PASCAL VOC (XML), YOLO (TXT), and CreateML annotation formats and can be deployed on Windows, macOS, and Linux operating systems. LabelMe (Wada et al., 2021) supports six different bounding box shapes, including polygon, rectangle, circle, line, point, and line strip. One of its limitations is that object labels
can only be saved and exported in JSON format. GTCreator (Bernal et al., 2019) allows multiple annotators to work simultaneously on the same task and offers full annotation editing and browsing capabilities. ByLabel (Qin et al., 2018) is a boundary-based semiautomatic tool that simplifies the labeling process by selecting among the boundary fragment proposals that the tool automatically generates. However, offline tools may cause compatibility issues with the operating system. Online tools allow data to be quickly annotated by enabling team collaboration. The VGG Image Annotator (VIA) tool (Dutta and
Zisserman, 2019) is open-source software that supports both offline and online annotation. Labels annotated in VIA can be exported to plain text data formats like JSON and CSV. The downside of the tool is that it lacks dataset management capabilities. ImageTagger (Fiedler et al., 2019) provides data and user management, manual and automatic labeling, annotations validation, and collaboration capabilities. Its annotations can be exported to a user-defined format. BRIMA



(Lahtinen et al., 2021) creates a browser-based extension to help researchers and crowdsourcing contributors conduct online
image annotation. Its annotation files can only be exported to the JSON format of MS COCO. Labelbox (Sharma et al., 2022)
provides many advanced features such as collaboration, automation, data and user management, and multiple format support.
Yet, its basic version can only realize the labeling of rectangular boxes and polygons.

## 2.3 Training data specifications

Using a unified or common way to describe labels is essential to facilitate training data sharing (Geetha et al., 2021). Common
data formats for object detection tasks mainly include the JSON format adopted by the Microsoft COCO (Lin et al., 2014)
dataset, the XML format adopted by the PASCAL Visual Object Classes VOC (Everingham et al., 2015) dataset, and the text
format adopted by models of YOLO (Redmon et al., 2016) series. In COCO, a JSON annotation is created for training, testing,
and validation on the entire dataset. The unique bounding box is represented by the coordinates of the upper left corner, and
the width and height of the bounding box. Its format can be described as [x, y, w, h]. In Pascal VOC, an XML annotation is
created for each image in the dataset. The "size" keyword is used to store the size information of the corresponding image and
the "name" keyword is used to store the category of the object. The upper-left corner and lower-right corner coordinates are
used to represent the unique bounding box. Its format can be described as [$x_{min}$, $y_{min}$, $x_{max}$, $y_{max}$]. In YOLO, an annotation in
TXT format is created for each image in the dataset. Its format is [x, y, w, h], which indicates the centroid coordinates, width,
and height of the bounding box after normalization, respectively.

145        In addition, the Spatio Temporal Asset Catalog (STAC) provides a common language to describe a range of geospatial
information, representing a single spatiotemporal asset as a GeoJSON feature plus date-time and links. Its bounding boxes are
represented using either 2D or 3D geometries. Yue et al. (2022) propose a Training data Markup Language (TDML) for
producing Machine learning training data, which defines a UML model and encodings consistent with the OGC standards
baseline. It supports the exchange and retrieval of the geospatial machine learning training data in the Web environment, which
is consistent with the ubiquitous JSON/XML encoding on the Web. It preserves the basic properties in other common data
specifications, while providing more detailed metadata for formalizing the information model of training data. Datasets
generated based on these standard data specifications will be more easily adopted and consumed by deep learning researchers.

## 3 Data generation of FASDD

Considering the limitations of the existing fire datasets in terms of number and visual tasks, this research intends to build a
large-scale, multi-source, multi-resolution, scene-complex, and standardized flame and smoke detection dataset (FASDD),
which is suitable for different application fields and compatible with image classification and object detection tasks. Figure 1
illustrates the workflow of generating FASDD. It mainly includes *data collection, data preprocessing, data annotation,* and
*quality control*. Based on these operating processes, we generate a computer vision dataset (FASDD_CV) and a remote sensing
(FASDD_RS) dataset. The CV and RS datasets are randomly split into the training set, validation set, and test set according to



1/2, 1/3 and 1/6 ratio. Then we merge these two different types of datasets into a unitary catalogue (FASDD) by conflating

their training sets, validation sets, and test sets, respectively. The data generation processes are described in more detail in the

following sections.

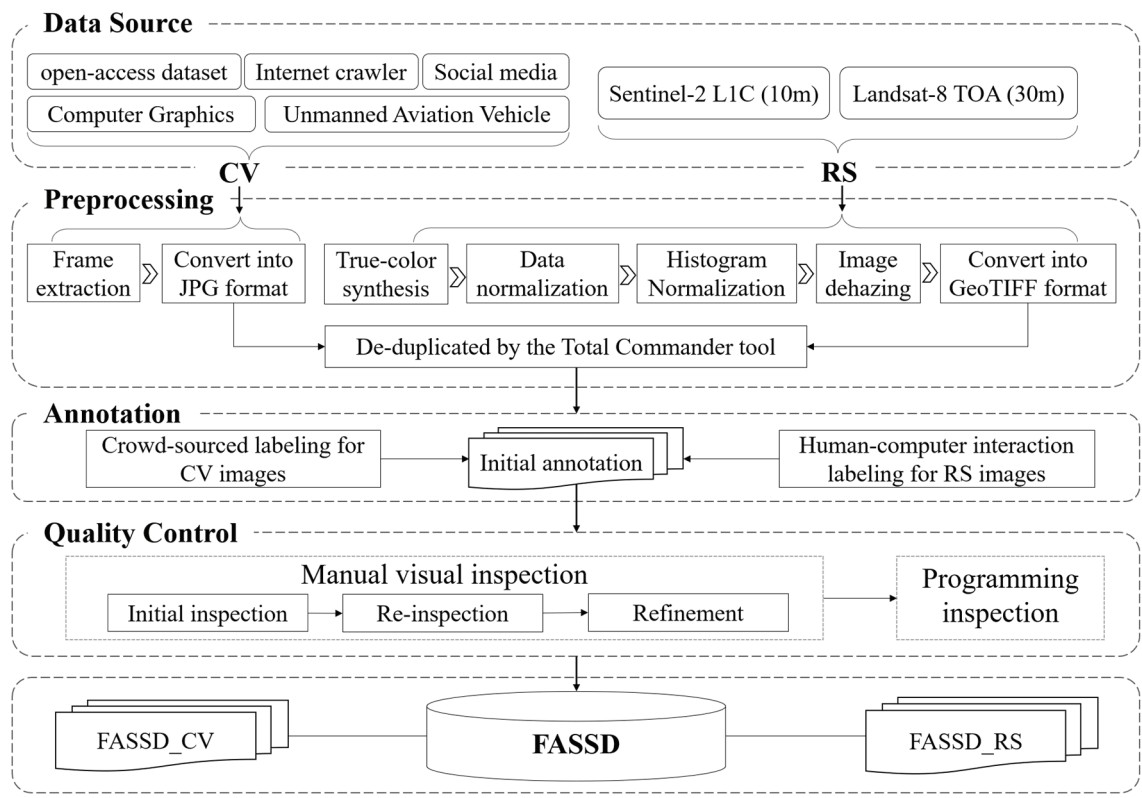

**Figure 1: The workflow for generating FASDD.**

**3.1 Data source**

To build a comprehensive fire dataset for CV tasks, various data sources are used, including existing open-access flame or

smoke datasets, social media images, CG (Computer Graphics) paintings, UAV images, and Internet crawler images. First, ten

available open flame or smoke datasets are reused, namely Wildfire Observers and Smoke Recognition Image and video

databases (Jakovcevic et al., 2010), Video-based smoke detection image database (Yuan, 2011), Wildfire smoke detection

datasets (Ko et al., 2012), the BoWFire (Best of both Worlds Fire detection) dataset (Chino et al., 2015), MIVIA database

(Foggia et al., 2015), Fire Detection Image Dataset (Sharma et al., 2017), Smoke Detection Datasets (Zhang et al., 2018), Fire

Image Data Set for Dunnings 2018 study (Dunnings et al., 2018), Fire-Smoke-Detection-Dataset (Geng et al., 2020) and the

FLAME (Fire Luminosity Airborne-based Machine learning Evaluation) dataset (Shamsoshoara et al., 2021). A large number

of source data in these datasets including CG images, UAV images, and video frames with good quality, are filtered, extracted,

and labeled. Some fire-related images are extracted from social media platforms such as TikTok. Objects easily confused with



smoke (e.g. dark clouds, shadows, hair, and impervious surfaces) and flame (e.g. lights, sunset glow, and reflective clothing) are considered as negative samples. Images about these negative samples are obtained on the Internet via the web crawler.

To produce representative samples of wildfires, ten typical areas (Hu et al., 2021) where wildfires have occurred frequently in recent years are selected (shown in Fig. 2). These regions cover all continents except Antarctica, including Canada, America,
Brazil, and Bolivia, Greece and Bulgaria, South Africa, China, Russia, and Australia. Satellite imagery of these regions captured during fire events is collected from Sentinel-2 with 10m resolution and Landsat-8 with 30m resolution. Sentinel-2 (Hu et al., 2021; Gargiulo et al., 2019) and Landsat-8 (De Almeida Pereira et al., 2021. Rostami et al. 2022) data have been adopted by many fire detection studies. Table 1 summarizes the details of remote sensing data sources used in this research. Since the atmospheric correction process may lead to the problem of missing pixels around the smoke and clouds in surface
reflection imagery, we make use of Sentinel-2 and Landsat-8 data products that are not corrected for atmospheric conditions, namely Sentinel-2 L1C (Level-1C) and Landsat-8 TOA (top-of-atmosphere), to generate FASDD_RS. A total of 310,280 remote sensing images with cloudy pixel percentages below 5% are collected. The RS image sizes range from $1000\times1000$ to $2200\times2200$.

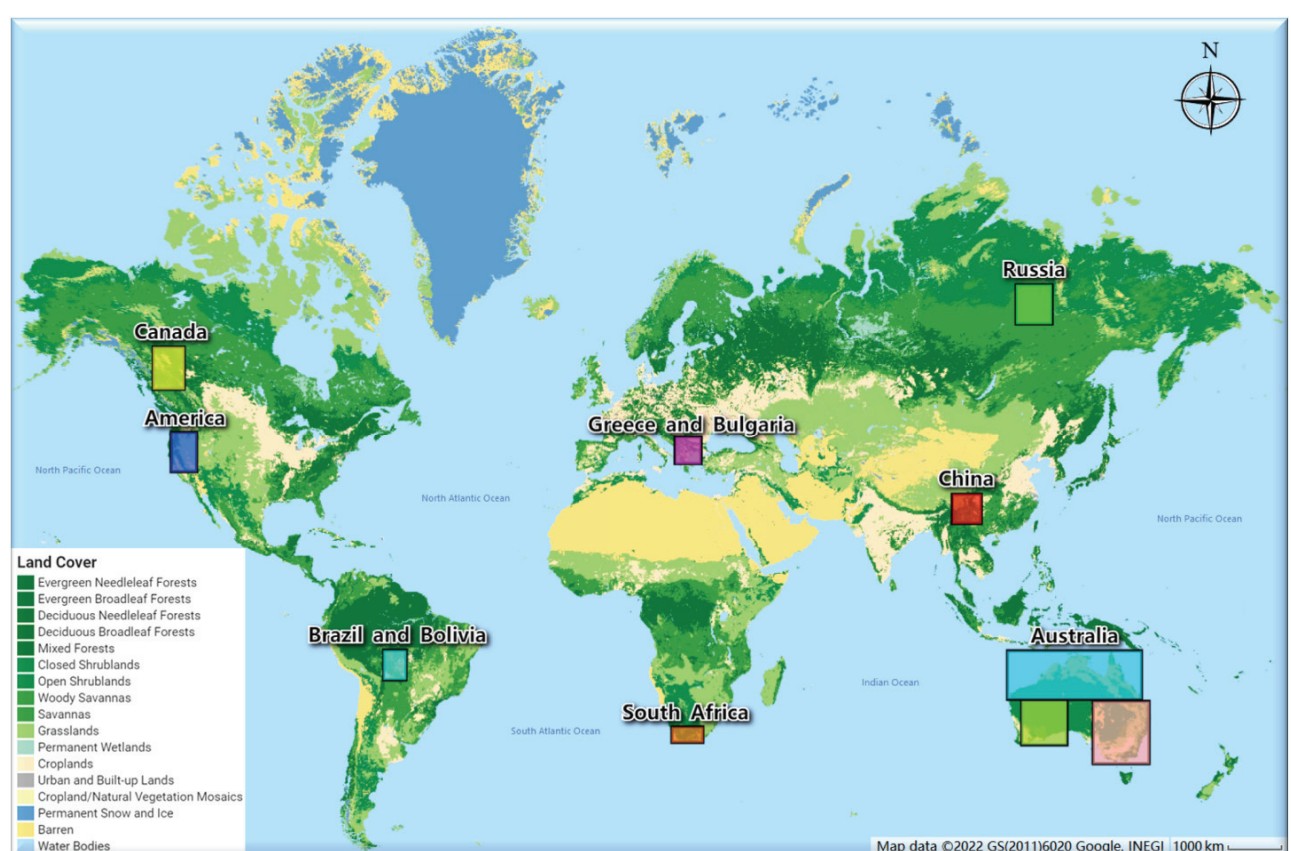

**Figure 2: The typical areas of fire events around the world. The base map (map data from Google Earth Engine © Google Services 2022) shows MODIS global land cover types at yearly intervals (Friedl and Sulla-Menashe, 2020) distributed by NASA's Land Processes Distributed Active Archive Center.**

**Table 1 The details of data collection for typical fire events around the world**

| Region | Continent | Time range | Spatial range | Data source | Number | Resolution |
|---|---|---|---|---|---|---|
| Canada | North America | 2018.08.05 - 2018.08.15 | [-129.00, 58.90], [-129.00, 53.06], [-120.08, 53.06], [-120.08, 58.90] | Sentinel-2, L1C | 5764 | 10m |
| America | North America | 2018.11.05 - 2018.11.15 | [-123.50, 44.62], [-123.50, 37.37], [-118.16, 37.37], [-118.16, 44.62] | Sentinel-2, L1C | 8437 | 10m |
| Brazil and Bolivia | South America | 2019.08.15 - 2019.08.25 | [-62.30, -11.18], [-62.30, -18.49], [-58.68, -18.49], [-58.68, -11.18] | Sentinel-2, L1C | 6977 | 10m |
| Greece and Bulgaria | Europe | 2018.07.15 - 2018.07.25 | [19.65, 43.16], [19.65, 38.59], [25.08, 38.59], [25.08, 43.16] | Sentinel-2, L1C | 10725 | 10m |
| South Africa | Africa | 2018.10.20 - 2018.10.30 | [18.76, -31.84], [18.76, -34.58], [25.92, -34.58], [25.92, -31.84] | Sentinel-2, L1C | 9573 | 10m |
| China | Asia | 2020.03.30 - 2020.04.05 | [101.28, 28.25], [101.28, 27.84], [101.65, 27.84], [101.65, 28.25] | Sentinel-2, L1C | 624 | 10m |
| Russia | Europe | 2018.07.15 - 2018.07.25 | [118.05, 66.69], [118.05, 64.81], [122.26, 64.81], [122.26, 66.69] | Sentinel-2, L1C | 2111 | 10m |
| Australia | Oceania | 2019.07.01 - 2020.02.20 | [113.10, -10.81], [113.10, -23.77], [151.16, -23.77], [151.16, -10.81] | Sentinel-2, L1C | 182932 | 10m |
| | | | [137.37, -23.68], [137.37, -38.99], [153.27, -38.99], [153.27, -23.68] | Landsat-8, TOA | 52669 | 30m |
| | | | [117.41, -23.84], [117.41, -34.41], [129.89, -34.41], [129.89, -23.84] | Landsat-8, TOA | 30468 | 30m |

## 3.2 Data preprocessing

To ensure the consistency and standardization of FASDD, some basic preprocessing steps on source data shall be conducted before data annotation. For video data, key frame extraction is performed and images are sampled in a step of 30 frames to ensure the difference between samples. Then all CV images (including images extracted from videos) are converted into JPEG format files. For remote sensing imagery, additional processing steps are required including true-color synthesis, data normalization, and image de-hazing. All remote sensing images are synthesized into true-color images for human interpretation. Pixel values are normalized to the range of 0-255. These preprocessing steps allow them to be suitable for general flame and smoke detection models. Then, histogram normalization and dehazing are performed to adjust the image color components and improve the image clarity. And all remote sensing images are saved as GeoTIFF format files. In the end, all images with



the same content are de-duplicated based on the Total Commander tool (Total Commander, 2022) to ensure the difference and
uniqueness of image features in the dataset.

### 3.3 Data annotation

For CV data annotation, all selected images are distributed to more than 70 volunteers in the field of fire detection for
collaborative labeling in a crowd-sourced manner. Volunteer annotators are asked to label flame and smoke objects in images
using non-directional minimum bounding rectangles. Although data are labeled offline with the LabelImg tool, some basic
annotation rules are formulated to standardize the labeling process. The annotation rules can be summarized as follows:

- A flame or a smoke object that is partially occluded but obviously connected is regarded as a separate object;
- Multiple tiny objects clustered together are considered to be a particular object;
- Flame or smoke with significantly different colors are not considered to be the same object;
- Objects smaller than 10×10 pixels and without apparent flame or smoke characteristics are ignored;
- Reflections of flame and smoke on smooth surfaces such as water shall be ignored if they do not match the shape and
texture features of the corresponding original objects;
- Objects smaller than 10×10 pixels with prominent shape and texture features shall be not omitted;
- Images smaller than 48×48 and difficult to be interpreted shall be deleted.

For RS data annotation, we adopt a semi-automatic way to annotate RS images with human-computer interaction. First,
target images that may contain flame and smoke objects or confusing objects are manually searched and screened. All the
target images are distributed to a small group of trained annotators to produce positive samples. Meanwhile, a flame and smoke
detection model trained on existing FASDD_CV is employed to predict semantic tags of target images. Those images with
confidence greater than 80% are further screened out from the inference results, and labels in those images similar to flame
and smoke are manually annotated as negative samples. In the annotation process, spatial information of all remote sensing
images, including longitude, latitude, and projection information, is retained for the localization and tracking of forest fire
events. Finally, 5,773 images are annotated based on human-computer interaction.

The flame and smoke objects in FASDD are given the labels "fire" and "smoke" for the object detection task, respectively.
Annotation files in four kinds of formats are provided in FASDD, i.e., the JSON format defined by the TDML (Yue et al.,
2022), the XML format adopted by the PASCAL VOC (Everingham et al., 2015) dataset, the JSON format adopted by the
Microsoft COCO (Lin et al., 2014) dataset, and the text format adopted by models of YOLO (Redmon et al., 2016) series.
Examples of four annotation formats are displayed in the attached file. Since all images could be classified into four semantic
categories, i.e. "Fire", "Smoke", "FireAndSmoke", and "NeitherFireNorSmoke", the category label is added to each image
filename as the prefix. With such category prefixes, FASDD could also be used to train fire scene classification models.



### 3.4 Quality control

To ensure the quality of the dataset, we develop a set of quality control procedures shown in Fig. 3. On the one hand, three-stage manual visual inspection procedures are designed after obtaining the initial annotation files of the dataset, i.e., initial inspection, re-inspection, and refinement, to correct unconfident data. In the *initial inspection* stage, every two annotators are assigned to one group to cross-check and modify the annotation files against each other, which helps find out inconsistent labels between different interpreters and reduces cognitive biases in crowdsourcing annotations. In the *re-inspection* stage, a 240 small group of quality inspectors is trained to audit the results from the initial inspection stage to eliminate omission errors and fine-tune the position, category, width, and height of bounding boxes. In the *refinement* stage, we invite well-trained domain experts to resolve annotation conflicts from the initial stage and relabel difficult-to-determine labeling cases from the previous stages. On the other hand, we introduce a *programming inspection* procedure after manual visual inspection procedures. The programming inspection procedure performs final data cleaning on annotation files using annotation checking 245 code. The code could automatically modify empty, duplicated, or range overflow bounding boxes, and misclassified or misspelled labels to prevent invalid and omitted values that are not easily detectable by humans. After these inspection steps, the consistency and standardization of annotation files can be ensured as much as possible.

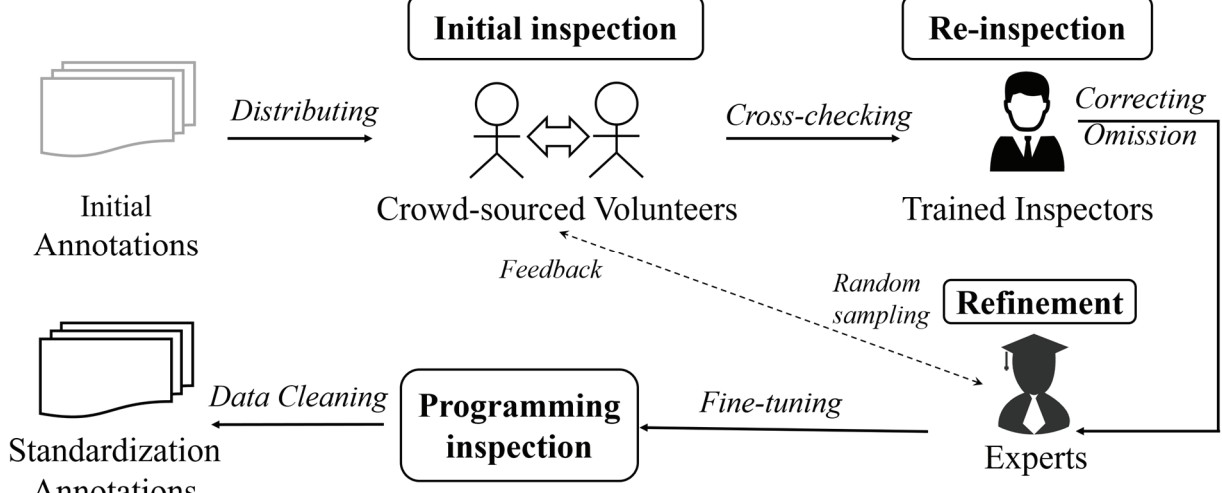

**Figure 3: The quality control flowchart**

### 3.5 Dataset characteristics and values


FASDD contains fire, smoke, and confusing non-fire/non-smoke images acquired at different distances (near and far), different scenes (indoor and outdoor), different light intensities (day and night), and from various visual sensors (surveillance cameras, UAV, and satellites). FASDD consists of two sub-datasets, a CV dataset (i.e. FASDD_CV) and an RS dataset (i.e. FASDD_RS). It is worth mentioning that the CV dataset and RS dataset could be used individually. The reason for merging the two datasets into a unitary catalog is twofold. First, we hope that the large diversity of training data in CV and RS domains



could help build a large fire detection model with improved performance and generalization capability (for some scenarios at least). Second, models trained on FASDD could be simultaneously deployed in spaceborne, airborne, and terrestrial sensors to build a space-air-ground integrated fire detection network.

**Table 2: The composition and characteristics of the FASDD**

| Datasets | Images | Positive samples | Negative samples | Flame objects | Smoke objects | Width range | Height range | Aspect ratios |
|---|---|---|---|---|---|---|---|---|
| FASDD_CV | 95,314 | 56,115 | 39,199 | 73,297 | 53,080 | 78~10,600 | 68~8,858 | 1:6.6~1:0.18 |
| FASDD_RS | 5,773 | 3,062 | 2,711 | 9,369 | 4,662 | about 1,000×1,000 or 2,200×2,200 | about 1,000×1,000 or 2,200×2,200 | 1:3.05~ 1:0.85 |
| FASDD | 101,087 | 59,177 | 41,910 | 82,666 | 57,742 | 78~10,600 | 68~8,858 | 1:6.6~1:0.18 |

Table 2 lists the composition and characteristics of the FASDD. A total of 101,087 samples are produced, of which 59,177 are annotated as positive samples, and 41,910 are labeled as negative samples. Some example images of FASDD are shown in Fig. 4. There are 82,666 flame object instances and 57,742 smoke object instances labeled in the entire dataset. For the sub-datasets, FASDD_CV consists of 95,314 general computer vision (CV) samples, and FASDD_RS consists of 5,773 remote sensing (RS) samples. FASDD_CV contains 73,297 fire instances and 53,080 smoke instances. The size of CV images spans
a relatively large range, with a width range of 78~10,600 pixels and a height range of 68~8,858 pixels. The image aspect ratios are also quite different, widely ranging from 1:6.6 to 1:0.18. FASDD_RS contains 9,369 fire instances and 4,662 smoke instances. The sizes of remote sensing images are mainly distributed around 1,000×1,000 or 2,200×2,200 pixels.

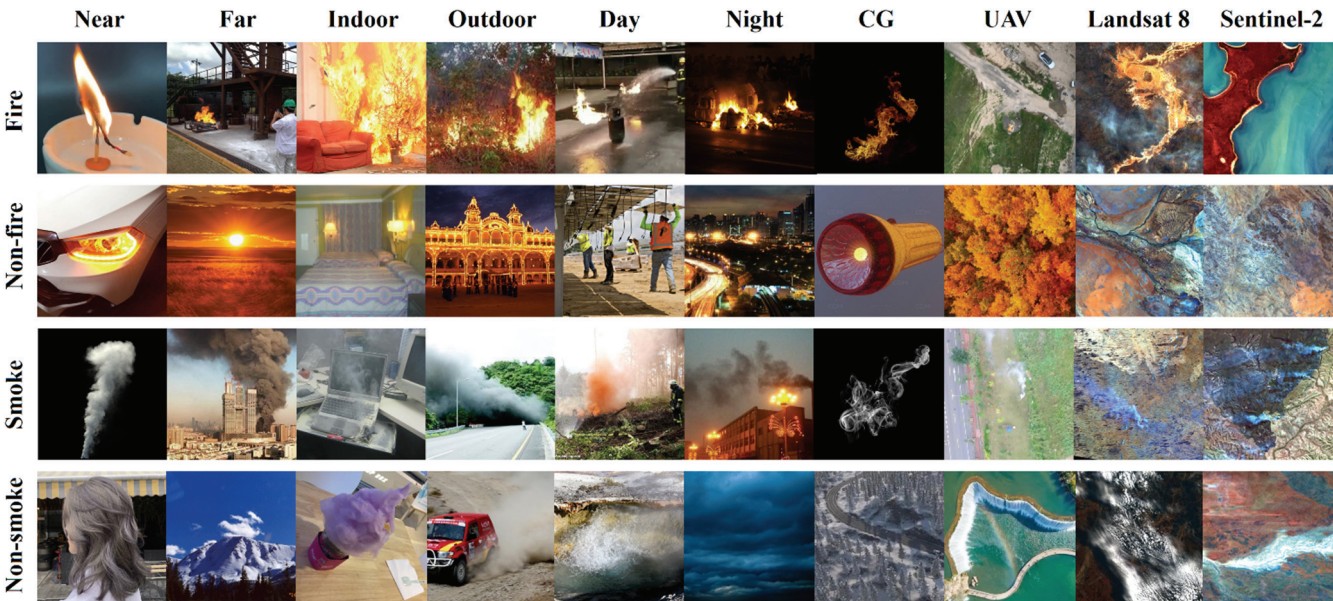

**Figure 4: The example images in FASDD. CV images are from open-access datasets (Chino et al., 2015; Sharma et al., 2017; Geng**
**et al., 2020). RS images are from Landsat-8 TOA and Sentinel-2 L1C.**



Compared with existing fire datasets including FLAME (Shamsoshoara et al., 2021), MIVIA (Foggia et al., 2015), and BoWFire (Chino et al., 2015), FASDD has the following remarkable characteristics.

(1) Large Scale. FASDD consists of more than 100,000 images and 140,000 object instances that are manually labeled with bounding boxes. To the best of our knowledge, it is the most versatile, comprehensive, and publicly available dataset for fire detection.

(2) Rich sample variations. The proposed FASDD dataset holds rich variations in image size, resolution, illumination (day and night), scene (indoor and outdoor), image range (far and near), sensor (surveillance cameras, UAV sensors, and satellite), and data source (Internet, social media, and open-access fire datasets). Such image variations will help enhance the robustness of models.

(3) High intra-class diversity and some inter-class similarity. Due to the characteristics of growth, disorder, color diversity, and intensity variability of flame and smoke, objects in the same category have different sizes, postures, and colors. There are also some similarities between flame and smoke, such as the red smoke by the glow of flame looks like the flame.

(4) Small objects of flame and smoke. It is well known that small object detection is a challenging problem in deep learning research related to computer vision. FASDD contains a large number of small flame and smoke objects, especially flame objects from remote sensing imagery and far-field wildfire images.

(5) Geo-referenced images. Compared with traditional CV datasets, FASDD contains many georeferenced images. The location information in remote sensing images can be used to detect or infer the location of fire events in time.

## 4 Evaluation and application

### 4.1 Experiment Setup

In our experiment, we randomly select half of the dataset for training, 1/3 for validation, and 1/6 for testing following existing study (Agrawal et al., 2014; Xia et al., 2018; Ma et al., 2022). Four classical models with significant architectural differences are selected for performance evaluation, including the two-stage Faster-RCNN (Ren et al., 2015), the one-stage anchor-free GFL (Li et al., 2020), the anchor-based YOLOv5x (Jocher et al., 2021), and the Swin Transformer (Liu et al., 2021) that achieves state-of-the-art (SOTA) performance on the COCO dataset. We use the same training configuration for all models participating in the evaluation to ensure the fairness of performance comparison. For YOLOv5, we use an SGD optimizer with a learning rate of 0.02, a momentum of 0.9, and a weight_decay of 0.0001. For Faster RCNN, GFL, and Swin Transformer, we use an SGD optimizer with a learning rate of 0.02, a momentum of 0.937, and a weight_decay of 0.0005. In addition, We use data augmentation, batch normalization, early stopping, and warm-up to prevent overfitting. All models are trained from scratch without using pre-trained weight files. The only exception is that YOLOv5x uses an image size of 960×960, while other models use an image size from 1333×480 to 1333×800. The original images will be resized to the above size in training and inference processes, which avoids the limitation of the model to images with different spatial resolutions. Other parameters





are consistent for the four models, including epoch 36 and batch size 2. In terms of GPU devices, all models for are trained, validated and tested on an NVIDIA GeForce RTX 3090 with 24GB memory.

## 4.2 Evaluation metrics

Four metrics are used to quantitatively evaluate the accuracy of the model prediction results, including Precision, Recall, AP (Average Precision), and mAP (mean Average Precision). Precision represents the ratio of the correct prediction box to all prediction boxes. Recall represents the ratio of the correct prediction box to all ground-truth boxes. AP represents the area under the curve (AUC) of Precision-Recall for each class in the dataset. mAP represents the AP mean value of all classes. We select the mAP@0.5, a more representative mAP indicator, as the primary reference metric of model accuracy. The mAP@0.5

refers to mAP when the IoU (Intersection over Union) between prediction and ground-truth boxes is not less than 50%, which is usually used to evaluate the overall performance of models. Precision and Recall are calculated as shown in Eq. (1) and (2):

$$Precision = \frac{TP}{TP + FP} \tag{1}$$

$$Recall = \frac{TP}{TP + FN} \tag{2}$$

where, $TP$ represents the number of prediction boxes when the IoU between prediction and ground-truth boxes is not less than 0.5. $FP$ represents the number of prediction boxes when the IoU between the prediction and ground-truth boxes is less than 0.5. $FN$ represents the number of ground-truth boxes missed from detection.

## 315 4.3 Performance evaluation

We train Faster-RCNN, GFL, Swin Transformer, and YOLOv5x models on FASDD_CV, FASDD_RS, and FASDD, and evaluate and compare the accuracy of these classic object detection models based on the validation set and test set in the above three datasets, respectively. Table 3 shows the accuracy evaluation results of classical models on FASDD_CV, FASDD_RS and FASDD.

In the evaluation results on FASDD_CV, the overall accuracy of Faster-RCNN, GFL, YOLOv5x, and Swin Transformer is gradually increasing. Among them, the Faster-RCNN model achieves only 52.55% validation mAP@0.5 and 61.26% test mAP@0.5 on FASDD_CV. The GFL and Swin Transformer models exhibit good performance, and the Swin model achieves the highest validation accuracy of 74.60% on the $AP_{smoke}$ metric. The YOLOv5x model shows the best performance, achieving the highest accuracy in all metrics except $AP_{smoke}$, particularly the 84.07% mAP@0.5 accuracy on the test set. Considering the

evaluation results again, the worst-performing Faster-RCNN also achieves an evaluation accuracy higher than 60% on FASDD_CV, which is partly due to the contribution of the large-scale sample of FASDD_CV.

In the evaluation results on FASDD_RS, the overall accuracy of all models is significantly lower than that on FASDD_CV, which demonstrates the difficulty of fire detection on Remote Sensing images. Among them, the Faster-RCNN exhibits the lowest model performance. Compared with the Faster-RCNN model, GFL exhibits performance improvement to some extent.



And YOLOv5x outperforms the GFL on both the validation and test sets. The Swin Transformer model achieves the best performance with 56.70% and 53.01% mAP@0.5 on the validation and test set, respectively. This may be attributed to its transformer structure, which is better at capturing global contextual information and large-scale spatial relationships. Some smoke areas in remote sensing images of large fire are relatively large, so the Swin Transformer model achieves the most advanced performance on FASDD, especially on the $AP_{smoke}$ metric. In addition, we added ablation experiments to demonstrate

that model evolution can further improve the model accuracy on the RS dataset. Four models show different degrees of accuracy improvement by simply increasing the epoch to 72. Particularly, the YOLOv5x improves the mAP accuracy of the test set by 3.03%. In the foreseeable future, through careful fine-tuning of hyperparameters and continuous evolution of the algorithms, the fire detection models for object detection tasks can show more satisfactory performance on FASDD_RS.

**Table 3: Accuracy evaluation of classic object detection models**

| Datasets | Method | Epoch | Validation | | | Test | | |
|---|---|---|---|---|---|---|---|---|
| | | | $AP_{fire}$ (%) | $AP_{smoke}$ (%) | mAP@0.5 (%) | $AP_{fire}$ (%) | $AP_{smoke}$ (%) | mAP@0.5 (%) |
| FASDD_CV | Faster-RCNN | 36 | 48.20 | 56.90 | 52.55 | 67.40 | 55.20 | 61.26 |
| | GFL | 36 | 56.60 | 69.00 | 62.82 | 72.70 | 73.10 | 72.90 |
| | Swin Transformer | 36 | 65.00 | **74.60** | 69.79 | 81.50 | 76.10 | 78.77 |
| | **YOLOv5x** | 36 | **70.96** | 73.29 | **72.13** | **86.48** | **81.66** | **84.07** |
| FASDD_RS | Faster-RCNN | 36 | 25.80 | 31.60 | 28.66 | 24.3 | 39.8 | 32.05 |
| | GFL | 36 | 34.00 | 36.90 | 35.46 | 33.5 | 46.6 | 40.08 |
| | **Swin Transformer** | 36 | **43.40** | **56.80** | **50.10** | **41.00** | **65.00** | **53.01** |
| | YOLOv5x | 36 | 37.35 | 45.93 | 41.64 | 33.42 | 49.35 | 41.39 |
| FASDD_RS | Faster-RCNN | 72 | 26.40 | 34.00 | 30.22 | 24.60 | 37.90 | 31.24 |
| | GFL | 72 | 35.30 | 38.80 | 37.06 | 33.60 | 47.70 | 40.63 |
| | **Swin Transformer** | 72 | 41.50 | **60.80** | **51.20** | 39.50 | **67.20** | **53.34** |
| | YOLOv5x | 72 | **48.36** | 49.74 | 45.97 | 34.40 | 54.45 | 44.42 |
| FASDD | Faster-RCNN | 36 | 44.30 | 53.70 | 49.00 | 59.00 | 51.50 | 55.24 |
| | GFL | 36 | 53.60 | 67.10 | 60.35 | 65.80 | 70.60 | 68.20 |
| | Swin Transformer | 36 | 58.90 | **72.20** | 65.55 | 71.20 | 71.20 | 73.20 |
| | **YOLOv5x** | 36 | **67.80** | 72.10 | **69.94** | **79.14** | **79.17** | **79.15** |

In the evaluation results on FASDD, the two-stage object detection model, Faster-RCNN, shows the lowest performance on both FASDD validation and test set with 49.00% and 55.24% mAP@0.5 respectively. Compared with Faster-RCNN, the one-stage anchor-free GFL model obtains 11.35% and 12.96% mAP@0.5 performance gains on validation (60.35%) and test (68.20%) sets. Compared with Faster-RCNN and GFL, Swin Transformer has significant performance improvement. Moreover, its validation set accuracy (65.55%) and test set accuracy (73.20%) are the closest to the accuracy evaluation results

of YOLOv5x, showing a pretty competitive accuracy evaluation result. The anchor-based YOLOv5x exhibits state-of-the-art performance on FASDD, achieving the highest 69.94% validation accuracy and 79.15% test accuracy on the mAP@0.5 metric.



The accuracy of the Swin Transformer is slightly lower than that of YOLOv5x. The reason may be that the parameter configuration and training strategy of the two models is only as consistent as possible but not entirely consistent, which may lead to a loss of comparability to some extent.

Experiments show that the detection accuracy of the classical object detection model on FASDD_CV is generally better than that on FASDD_RS. In terms of overall assessment results, the models also demonstrate good detection performance on FASDD that integrates cross-domain data (CV and RS). This indicates that the pre-trained model trained on FASDD can achieve good accuracy, generalizability, and transfer learning capability on the cross-domain object detection task. However, FASDD is still challenging and there is sufficient space to improve its detection accuracy. It can be used to assist researchers

in developing more targeted and robust algorithms to promote new developments in fire detection. Moreover, based on FASDD, we can provide pre-trained large models with better generalization performance for downstream tasks such as object detection and semantic segmentation.

In addition, to validate the generalization of models driven by cross-domain FASDD, we evaluate the inference accuracy on FASDD_CV and FASDD_RS using models trained on FASDD. Table 4 shows the evaluation results. From the evaluation

results on FASDD_CV, all models trained on FASDD demonstrate a similar performance compared with these models trained on FASDD_CV only. From the evaluation results on FASDD_RS, compared with all models trained on FASDD_RS only, Faster-RCNN, GFL, and Swin Transformer show an accuracy loss of 3.4% to 9.1%. However, in contrast to the above models, YOLOv5x obtains an accuracy gain of 5.6%, which gains significant accuracy improvement on FASDD_RS while maintaining accuracy on FASDD_CV.

**Table 4: Accuracy evaluation on FASDD_CV and FASDD_RS using models trained on FASDD**

| Datasets | Method | Test | | |
|---|---|---|---|---|
| | | $AP_{fire}$ (%) | $AP_{smoke}$ (%) | mAP@0.5 (%) |
| FASDD_CV | Faster-RCNN | 66.40 | 54.40 | 60.37 |
| | GFL | 72.40 | 74.10 | 73.24 |
| | Swin Transformer | 78.60 | 77.30 | 77.95 |
| | **YOLOv5x** | **86.54** | **81.32** | **83.93** |
| FASDD_RS | Faster-RCNN | 19.90 | 29.10 | 24.46 |
| | GFL | 31.10 | 42.30 | 36.68 |
| | Swin Transformer | 30.20 | 57.60 | 43.90 |
| | **YOLOv5x** | **34.93** | **59.09** | **47.01** |

In summary, for a single fire detection task in the CV or RS domain, we recommend using only the FASDD_CV or FASDD_RS datasets. Moreover, experiments show that the best results can be achieved by using the FASDD_CV dataset with the YOLOv5x model and the FASDD_RS dataset with the Swin Transformer model. When faced with an integrated fire detection task across CV and RS domains such as a space-air-ground sensor network, we can use FASDD to train a common



DL model for spaceborne, airborne, and terrestrial sensors simultaneously. Meanwhile, we recommend using the YOLOv5x
model to train FASDD for better generalization performance.

**Figure 5: The visual result of classical object detection models on the example images. Red circles indicate omitted flame, and yellow circles indicate omitted smoke. "Both" represents images of the "FireAndSmoke" category, and "None" means images of the "NetherFireNorSmoke" category. CV images are from open-access datasets (Chino et al., 2015; Dunnings et al., 2018; Geng et al., 2020). RS images are from Landsat-8 TOA and Sentinel-2 L1C.**



## 4.4 Visual results

Figure 5 shows the prediction results of classic object detection models on FASDD_CV and FASDD_RS example images, and compares them on four categories of fire images, i.e. Fire, Smoke, FireAndSmoke, and NetherFireNorSmoke. In the
CV_Fire results, Faster-RCNN incorrectly detects a gold necklace as flame and a black shoe as smoke. In the CV_Smoke results, the lights on the fire truck and helmets of firefighters bring challenges to Faster-RCNN and Swin Transformer. In the CV_Both results, Faster-RCNN incorrectly detects grey shadows on the ground as smoke. In the CV_None results, Faster-RCNN, GFL, and SwinTransformer incorrectly detect colored parrots as flame, and Faster-RCNN detects black backgrounds as smoke. In the RS_Fire results, all models have different degrees of omission errors. In particular, GFL does not detect the
existence of flame objects at all, and Faster-RCNN incorrectly detects large areas of water as smoke. In terms of the RS_Smoke, Faster-RCNN, and GFL show obvious problems of missed detection. In RS_Both, all models show missed detection of flame and smoke objects, and the missed detection of Faster-RCNN is severe. In the RS_None category, only the Faster-RCNN model incorrectly detects the dark blue surface as smoke, and none of the other models shows the false alarm. To sum up, in terms of image features, the significance level of flame and smoke features in FASDD_RS images is slightly less than that of
FASDD_CV images, and the flame in FASDD_RS image is easily confused with remote sensing ground objects or various scenes in reality. That is to say, detecting flame in remote sensing images is much more complex and challenging than in CV images.

    In terms of model performance, the false alarm rate of Faster-RCNN is higher than other models, and the model has the worst performance. The notable feature of GFL is its highly-missed detection rate on FASDD_RS. Swin Transformer also
shows the false alarm and missed detection, yet the overall detection effect is good. YOLOv5x can achieve a satisfactory detection effect except for a few missed detections on FASDD_RS. These results are obtained under the small batch size and epoch training configuration. Better detection results are possibly achieved using the optimized algorithms, the tuned parameters, or an extended training period.

## 4.5 Application in wildfire location

We apply the above classical methods to fire localization experiments in wildfire scenarios from remote sensing images. The latitude and longitude coordinates of the predicted boxes are used to verify the positioning accuracy of these methods. First, the coordinate system of the RS_Smoke image in Fig. 5 is converted to WGS84-based GPS coordinates. Then, inferences are performed on the georeferenced RS_Smoke image with four trained object detection models respectively. Finally, the positions of all prediction boxes are converted into the form of latitude and longitude coordinates.

Table 5 compares the geographic coordinates (top left and bottom right), centroid distance bias, and IoU between the prediction boxes and ground truth boxes of the RS_smoke image. "-" indicates the missed detection boxes, and the redundant boxes of false alarm are not added to the table. Among them, GFL misses three bounding boxes, Faster-RCNN misses two bounding boxes, Swin Transformer misses one bounding box, and YOLOv5x can detect all the bounding boxes. Compared



with the detection results of other models, the predicted geographic coordinates of the YOLOv5x model and the ground-truth
boxes are closer, showing a good fire site localization effect. In terms of the centroid distance, the prediction box centroids of
both Faster-RCNN and GFL are relatively far from the ground truth centroids, and the prediction box centroids of Swin
Transformer and YOLOv5x are relatively closer to the ground truth centroids. In terms of the IoU, Swin Transformer, and
YOLOv5x also exhibit good results around 90% on most of the prediction boxes. In particular, YOLOv5x achieves the highest
IoU between all prediction boxes and ground truth boxes. The above comparison results show that the YOLOv5x model trained
on FASDD helps to accurately locate and track wildfire sites in remote sensing images. This has practical significance for
detecting and monitoring large-scale forest fires using in-orbit satellites.

**Table 5: Comparison of coordinates between prediction and ground truth boxes**

| Box | Model | Top Left Coordinate | Bottom Right Coordinate | Centroid Distance Bias (m) | IoU |
|-----|-------|---------------------|-------------------------|----------------------------|-----|
| | Faster-RCNN | [142.5292, -13.8503] | [142.5413, -13.8578] | 452.22 | 0.33 |
| | GFL | - | - | - | - |
| Box1 | Swin Transformer | - | - | - | - |
| | **YOLOv5x** | **[142.5283, -13.8507]** | **[142.5405, -13.8564]** | **360.03** | **0.37** |
| | Ground Truth | [142.5251, -13.8453] | [142.5406, -13.8559] | 0.00 | 1.00 |
| | Faster-RCNN | [142.5826, -13.8343] | [142.5971, -13.8477] | 646.24 | 0.49 |
| | GFL | - | - | - | - |
| Box2 | Swin Transformer | [142.5831, -13.8347] | [142.6084, -13.8486] | **31.62** | 0.90 |
| | **YOLOv5x** | **[142.5836, -13.8341]** | **[142.6070, -13.8483]** | 63.25 | **0.93** |
| | Ground Truth | [142.5841, -13.8344] | [142.6076, -13.8483] | 0.00 | 1.00 |
| | Faster-RCNN | - | - | - | - |
| | GFL | - | - | - | - |
| Box3 | Swin Transformer | [142.4985, -13.8446] | [142.5161, -13.8555] | **36.06** | 0.82 |
| | **YOLOv5x** | **[142.4999, -13.8452]** | **[142.5159, -13.8555]** | 65.19 | **0.87** |
| | Ground Truth | [142.4997, -13.8448] | [142.5155, -13.8548] | 0.00 | 1.00 |
| | Faster-RCNN | - | - | - | - |
| | GFL | [142.4961, -13.8144] | [142.5223, -13.8535] | 917.40 | 0.52 |
| Box4 | Swin Transformer | [142.4965, -13.8104] | [142.5306, -13.8437] | 127.48 | 0.83 |
| | **YOLOv5x** | **[142.4976, -13.8126]** | **[142.5291, -13.8401]** | **68.01** | **0.89** |
| | Ground Truth | [142.4968, -13.8113] | [142.5288, -13.8410] | 0.00 | 1.00 |

## 5 Data availability

FASDD is freely available from the Science Data Bank website at https://doi.org/10.57760/sciencedb.j00104.00103 (Wang et
al., 2022a). There are a total of three compressed files, FASDD_CV.zip, FASDD_RS.zip, and FASDD.zip representing the
CV dataset, the RS dataset, and the full dataset composed of CV and RS respectively. Each zip file contains an "images" folder
for storing data and an "annotations" folder for storing labels. The "annotations" folder consists of label files in four formats:
VOC, COCO, YOLO, and TDML. In each format of labels, the dataset is randomly divided into training, validation, and test



sets with a ratio of 1/2, 1/3, and 1/6. The prefixes of image and label names are divided into "Fire", "Smoke", "FireAndSmoke",

and "NeitherFireNorSmoke", which represent different categories of data for classification tasks. The labels contain the classes "fire" and "smoke" to represent two common objects in fire images for object detection tasks. When faced with a single fire detection task in the CV or RS domain, we suggest using only FASDD_CV or FASDD_RS for model training. When faced with a fire detection task across CV and RS domains, we recommend FASDD to train a common DL model for both the CV and RS domains.

## 6 Cross-dataset validations


We chose the reliable Monitoring Trends in Burn Severity (MTBS) product (Finco et al., 2012) to validate the cross-dataset generalization (Torralba et al., 2011) of FASDD. This is because MTBS, with a higher spatial resolution (30m), is more convenient for fine-grained comparisons over local areas than MODIS fire products (e.g. from Giglio et al., 2018 and Giglio et al., 2016). Based on the multi-source data including Landsat from pre-fire and post-fire, MTBS maps the burn severity and

counts the burned area. Meanwhile, the Swin Transformer with the best performance on FASDD_RS is selected as the fire detector, and the cross-dataset validation is performed by comparing its prediction results with MTBS in the same fire scenario (Landsat-8 imagery from California, USA, 8 November 2018). The results as depicted in Fig. 6 show that our trained model can work well on the detection task of flame and smoke objects during fires. Compared to the MTBS product, our predictions demonstrate good cross-dataset validation results. It has a large area intersection with the mapped areas of MTBS, covering

almost all three different levels (Low, Moderate, and High) of burned areas in MTBS. When compared with the original imagery, it is also clear that we achieve satisfactory detection results for flame and smoke objects.

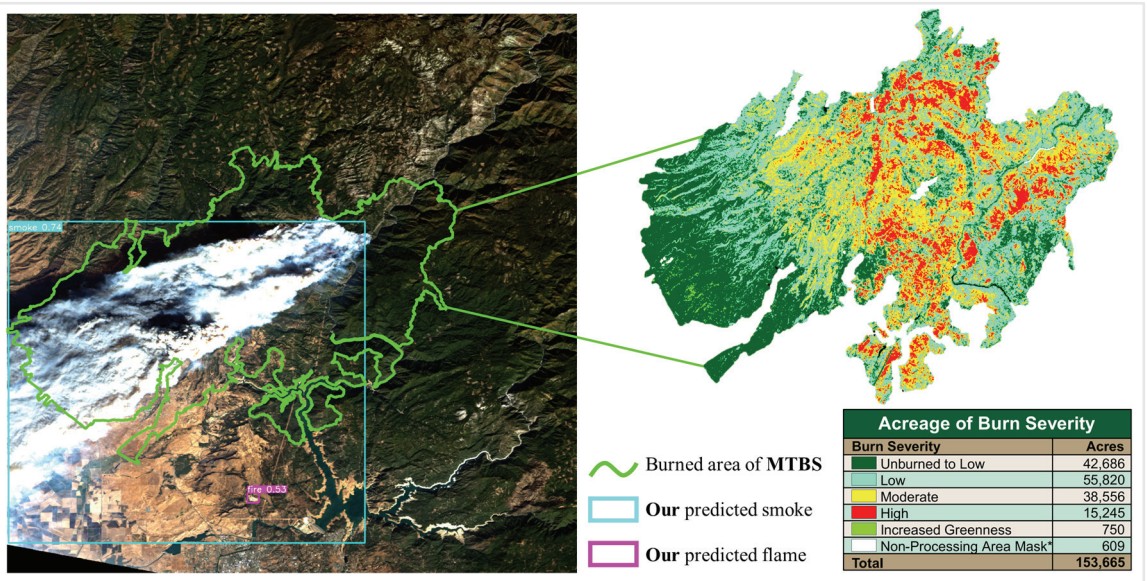

**Figure 6: Cross-dataset validation results for FASDD based on MTBS fire product.**

**7 Conclusion**

This paper presents an open-access 100,000-level Flame and Smoke Detection Dataset (FASDD). To the best of our knowledge, it is the largest fire detection dataset with the most variety of scenes, the highest heterogeneity, and the most significant difference in feature distribution. FASDD is compatible with image classification and object detection tasks. It provides four annotation files to enable out-of-the-box training samples for deep learning models. Especially, the use of TDML annotations provides a reference for the application of upcoming the OGC training data standard in the future. FASDD has significant

heterogeneity and challenges, laying a solid data foundation for future fire detection research.

  Based on the proposed dataset, we perform extensive performance evaluations and comparisons using multiple classic object detection models. The results show that the YOLOv5x model exhibits state-of-the-art performance with the highest test set accuracy close to 80%. That is to say, the trained YOLOv5x model can play a considerable role in the early warning and detection of urban fires or forest fires. Experiments demonstrate the merit of merging CV and RS datasets into a unique and

curated catalog. Models trained on FASDD achieve a similar performance compared with models trained on FASDD_CV. Yet, the YOLOv5x model trained on FASDD has better performance than that trained on FASDD_RS. The application in wildfire location also finds that the YOLOv5x model trained on FASDD can achieve high-quality location results.

  FASDD dataset provides a benchmark for developing advanced wildfire detection models that can be deployed on optical sensors mounted on watchtowers, drones, or satellites. Such models can be adapted to any other regional and global scale fire

scenarios, which can provide an important reference for government decision-making and fire rescue. Moreover, vision-based models trained on FASDD can also be combined with smoke sensors in practical applications for more accurate fire detection.

**Author contributions**

  MW, LJ, and PY conceived the study. MW wrote the first draft of the manuscript and managed data archiving. LJ and PY provided input on the overall methodology and participated in drafting the manuscript. DY and TT participated in the data

collection, data annotation, and quality control of the dataset. All authors discussed the results and commented on the manuscript.

**Competing interests**

  The authors declare that they have no conflict of interest.

**Acknowledgments**

We acknowledge Science Data Bank for publishing the dataset. We are grateful to the free access to the Landsat data provided by the USGS; the Sentinel data provided by the European Space Agency; the MCD12Q1 product provided by NASA's Land



Processes Distributed Active Archive Center, and the fire detection datasets provided by many researchers (Jakovcevic et al., 2010; Yuan, 2011; Ko et al., 2012; Chino et al., 2015; Foggia et al., 2015; Sharma et al., 2017; Zhang et al., 2018; Dunnings et al., 2018; Geng et al., 2020; Shamsoshoara et al., 2021). We would like to thank the Google Earth Engine team for sharing

the geospatial cloud platform, the Ultralytics team for sharing the YOLOv5 code (https://github.com/ultralytics/yolov5, last access: 16 November 2022), and the GitHub user K-H-Ismail for sharing Faster-RCNN, GFL and Swin transformer code (https://github.com/SwinTransformer/Swin-Transformer-Object-Detection, last access: 16 November 2022). Our thanks also go to the volunteer annotators who contributed to this dataset and the anonymous reviewers for their insightful suggestions.

**Financial support**

The work was supported by the National Natural Science Foundation of China (No. 42090011 and No. 42071354). Liangcun Jiang was also supported by the Fundamental Research Funds for the Central Universities (WUT:223108001).

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
