# Peer review of "FASDD: An Open-access 100,000-level Flame and Smoke Detection Dataset for Deep Learning in Fire Detection"

_Earth System Science Data, 2023_

## Referee Comment (RC2)

**Comments to the Authors**

This work constructs a 100000-level FASDD based on multi-source heterogeneous flame and smoke images, which provides a challenging benchmark to drive the continuous evolution of fire detection models. However, there are still some issues that need to be addressed to demonstrate the reliability of the dataset.

1.  In terms of sample annotation, what are the differences between satellite images with different spatial resolutions in the remote sensing field and RGB images in the CV field? Can different annotation formats be integrated or unified into one annotation format for the widespread use of FASDD?

2.  How to consider the difference in spatial resolution between Sentinel-2 L1C and Landsat-8 TOA? How to solve the detection accuracy of targets with different sizes? What are the differences in detection accuracy between large and small targets during sample annotation, model training, and inference stages?

3.  How to consider the difference in radiometric resolution between RGB and satellite images? How robust is the detection algorithm for data with different radiation resolutions? Please confirm through ablation experiments.

4.  The FASDD holds rich variations in image size, resolution, illumination, scenario, image range, viewing angle, platform, and data source. How to consider data at different scales for various deep learning models? Please choose the latest model in the CV field to verify the reliability of FASDD and the superiority of Transformer-based models, such as DETR, etc.

5.  What may be the reason for the poor validation performance of the transformer-based model compared to YOLOv5x?

6.  The annotation and partition ratio of samples directly affect the accuracy of deep learning models. Please supplement ablation experiments to demonstrate the reliability and universality of FASDD datasets under different training sample ratios.

7.  In the case of small samples, the Transformer-based model has poor convergence. The linear expansion of the transformer leads to a sharp increase in parameter size and insufficient local feature extraction. How to solve the problem of insufficient local feature extraction in transformer and poor target detection performance in FASDD?

8.  Is the low accuracy of the Transformer-based model caused by overfitting? Please provide the

training accuracy and validation accuracy curve of the Transformer-based model.

9. Please increase the types and quantity of deep learning algorithms (each model type contains at least two algorithms) to fully validate the universality and reliability of the dataset.

---

## Author Comment (AC5)

**RC1: 'Comment on essd-2023-73', Anonymous Referee #1, 17 Apr 2023**

**Summary**

Fire has wide impacts on Earth Systems and human society, and efficient fire detection could promote better understanding, modeling, and preventing fires. Wang et al., synthesized a comprehensive dataset (FASDD) covering images from terrestrial, airborne, and spaceborne sensors. For the previous version, I had nine major concerns about the data generation, data validation, and the usefulness of the data, but the current responses and version can hardly convince me for the majority of my major concerns. Here I reclaim my major concerns unaddressed and why they are critically important for fire detection using spaceborne satellite. I suggest the authors to handle those major concerns, otherwise, I really apologize and have to reject this work since these are critical or fundamental drawbacks for this study.

Response: Thank you for providing us with your insightful comments and feedback on our work. We greatly appreciate your expertise and the time you have dedicated to evaluating our paper. We have thoroughly revisited the concerns you raised in your previous review and have made significant improvements to address each of them. We understand the critical importance of data generation, data validation, and the usefulness of the dataset, particularly in the context of fire detection using satellite imagery. Our revisions aim to enhance the quality, reliability, and relevance of the dataset, aligning it more closely with the requirements of the study. We sincerely hope that the modifications we have made adequately address your concerns and demonstrate our commitment to providing a robust and valuable contribution to the field of fire detection. Once again, we express our gratitude for your feedback and assure you that we have taken your comments seriously in order to improve the overall quality of our work.

**Major comments**

(1) Data generation: it's well known that near infrared (NIR) and short-wave infrared (SWIR) are two commonly used bands for fire detection while the authors only used the visible bands via visual interpretation. If only based on true colors, it can hardly convince me about the generality of the dataset for large spatial scale fire detection (e.g., some land surface items could show similar colors with fires in remote sensing images and thus mislead machine leaning models). Also, the results in Table 3 showed the really low performance on detecting remote sensing fires even with advanced machine learning models. With such a lower accuracy, how can the data help improve fire detection.

My response to the authors' responses: I understand some previous works have used optical camera or RGB for smoke detection, but those purely RGB based fire detection works are used for terrestrial or near-surface fire monitoring instead of spaceborne satellite. The FASDD_RS data include sentinel-2 and landsat8, and the NIR or SWIR bands are fundamentally important for monitoring fire-induced vegetation and dryness changes, and are basically used for fire detection. For example, the listed works bellow all used bands like NRI or SWIR. Some basic indexes with NIR and SWIR can achieve substantially higher accuracy (e.g., Castillo et al., 2020) than the reported accuracy by this study. Also the deep learning work in Pereira et al.2021, achieved much higher accuracy than your work. I have never seen a published reliable fire dataset derived from satellite observations only used RGB.

Barboza Castillo, E., Turpo Cayo, E. Y., de Almeida, C. M., Salas López, R., Rojas Briceño, N. B., Silva López, J. O., ... & Espinoza-Villar, R. (2020). Monitoring wildfires in the northeastern peruvian amazon using landsat-8 and sentinel-2 imagery in the GEE platform. ISPRS International Journal of Geo-Information, 9(10), 564.

de Almeida Pereira, G. H., Fusioka, A. M., Nassu, B. T., & Minetto, R. (2021). Active fire detection in Landsat-8 imagery: A large-scale dataset and a deep-learning study. ISPRS Journal of Photogrammetry and Remote Sensing, 178, 171-186.

Hu, X., Ban, Y., & Nascetti, A. (2021). Sentinel-2 MSI data for active fire detection in major fire-prone biomes: A multi-criteria approach. International Journal of Applied Earth Observation and Geoinformation, 101, 102347.

Response: Thank you very much for your comments and suggestions. We acknowledge the importance of near-infrared (NIR) and short-wave infrared (SWIR) bands in fire detection, as they are widely utilized in satellite-based fire monitoring. These bands play a crucial role in various applications, such as evaluating burnt areas post-fire using spectral indices (Barboza Castillo et al., 2020) or extracting fire segmentation masks employing deep learning techniques (De Almeida Pereira et al., 2021). In response to your concerns, we have made significant revisions to our dataset.

Firstly, we have incorporated the SWIR bands into the FASDD_RS dataset by combining two SWIR (B12 and B11) bands and the Red (B4) band of Sentinel-2 imagery, following the methodology employed by Hu et al. (2021). This pseudo-color image dataset, FASDD_RS (SWIR), is specifically designed for flame object detection.

Additionally, we have developed FASDD_RS (RGB) to cater specifically to smoke object detection. This dataset utilizes the R, G, and B bands to enable accurate identification and analysis of smoke objects within the imagery.

Notably, we have retained all spectral bands, resulting in the creation of FASDD_RS (All Bands) for simultaneous detection of flame and smoke objects. By introducing these three datasets, FASDD_RS (All Bands), FASDD_RS (SWIR), and FASDD_RS (RGB), we aim to address the limitations of the initial RS dataset and provide more comprehensive options for flame and smoke detection.

We evaluated the performance of these revised datasets using the Swin Transformer (ST) model, and the results are presented in Table 3. The models trained on FASDD_RS (SWIR) and FASDD_RS (RGB) exhibited enhanced performance in flame and smoke

detection, respectively. For flame detection using FASDD_RS (SWIR), the ST model achieved an mAP@0.5 score of 81.7% and a Recall score of 95.4% on the test set. Similarly, for smoke detection using the FASDD_RS (RGB) dataset, the ST model achieved an mAP@0.5 score of 68.8% and a Recall score of 88.8% on the test set.

The evaluation results for FASDD_RS (All Bands) demonstrated that the ST model performed well in detecting both flame and smoke objects, albeit with a decrease in accuracy compared to the smoke detection model trained on FASDD_RS (RGB) and the flame detection model trained on FASDD_RS (SWIR). This indicates that the inclusion of multispectral data facilitates simultaneous flame and smoke detection by leveraging information from a wider range of spectral bands. However, it is important to note that utilizing the additional bands incurs higher computational and memory requirements, and may introduce potential accuracy trade-offs due to potential interference from redundant bands. Notably, further improvements in model performance were obtained through the "pre-training + fine-tuning" approach, reaffirming the valuable contribution of large-scale heterogeneous data to the performance of fire detection models.

It should be noted that different deep learning tasks employ specific evaluation metrics to assess accuracy. Semantic segmentation tasks, as demonstrated in the work of De Almeida Pereira et al. (2021), often utilize metrics like F-Score, precision (P), and recall (R). In contrast, object detection tasks primarily rely on metrics such as mAP@0.5 and Recall. Object detection tasks involve predicting the position, size, and category of bounding boxes, presenting unique challenges distinct from classification and segmentation tasks. The provision of this benchmark dataset aims to facilitate research in the field of object detection and foster the development of advanced models for fire detection.

(2) Data generation: for active fire detection, middle infrared and thermal bands are also important but were ignored.

My response to the authors' responses: Actually, landsat8 has thermal infrared bands. My concern is that why only use RGB for fire detection? The ultimate goal of this dataset is for improving fire detection instead of some technique details like only using one machine learning model with the same-format inputting dataset. If fusing different datasets can improve fire detection, why not use comprehensive models (e.g., different models) to fuse different datasets and finally achieve better fire detection (e.g., based on an ensemble of machine learning models)? If you use this dataset to develop one single model but throwing important bands information, I don't think it is a good idea.

Response: Thank you for your valuable comments. We appreciate your point regarding the importance of including middle infrared and thermal bands for active fire detection. While Landsat 8 does offer thermal infrared bands, we made the decision to exclude them from the revised version of our dataset due to certain limitations. The atmospheric-corrected data of Landsat 8 presents challenges, as the smoke appears

similar to green vegetation, and there are anomalous black pixels around clouds. As a result, we opted to utilize Sentinel-2 L2A (Level 2A) products to generate the FASDD_RS. Additionally, the spatial resolution of Landsat 8 is lower compared to Sentinel-2. Given the time constraints of this study, we plan to produce our own Level-2 Landsat 8 data in the future. This will allow us to expand our RS dataset and incorporate the necessary thermal infrared bands for enhanced fire detection.

We would like to highlight that the inclusion of RGB bands in our current dataset was a deliberate choice. We considered the potential application of the trained model to real-time fire detection using video satellites or CubeSats. For example, Azami et al. (2022) demonstrated the feasibility of active wildfire detection onboard the CubeSat using deep learning models. They trained these models for the wildfire image classification task using visible RS images captured by various satellite sensors. The availability of RGB bands in diverse satellite imagery sources makes them more readily accessible for practical implementation.

We greatly appreciate your suggestion regarding the utilization of different models and dataset fusion. Our primary objective in this study was to establish a benchmark dataset for the fire object detection task. As part of our evaluation, we selected the Swin Transformer as the baseline model in the revised manuscript, which has exhibited state-of-the-art performance on the CV benchmark dataset, specifically the COCO dataset. To gain a comprehensive understanding of the experimental settings, we kindly suggest referring to Section 4.1 of our revised paper, where you can find more detailed information. However, your idea of employing ensemble models and integrating multiple datasets is intriguing and worth exploring in future research endeavors. We will take your suggestion into consideration and explore the incorporation of different models and the fusion of diverse datasets to further enhance the accuracy of fire detection. Once again, we genuinely appreciate your insightful feedback.

(3) Data generation: for the fire detection, the authors used the top-of-atmosphere reflectance instead of atmospheric-corrected land surface reflectance which should be a problem. For example, if the smoke is white or grey, how to classify smoke versus clouds only through visual interpretation of true-colors? Meanwhile, the CV fire images should be obtained on land surface with a much higher spatial resolution (can be with sub-meter resolution in Fig. 4). Can such kinds of CV image trained machine learning models be directly used to large-scale remote sensing data without atmospheric-correction and with different spatial resolutions?

My response to the authors' responses: I understand that FASDD_CV have images from UAVs. But from the results in Table 4, it seems that for FASDD_RS, the model performance even dropped if we used FASDD (including FASDD_RS and FASDD_CV) for training the model. The dropped performance means that the knowledge derived from FASDD_CV can hardly be transferred or used for improving FASDD_RS. If so, why we should integrate those two heterogenous datasets?

Response: Thank you for your valuable insights. We appreciate your expertise in emphasizing the significance of atmospheric correction for accurate smoke detection. Based on your feedback, we have taken the necessary steps to address this concern by re-downloading Sentinel-2 Level-2A data. This ensures that the impact of clouds on smoke detection is minimized, leading to more reliable results.

Regarding the integration of FASDD_CV and FASDD_RS datasets, we understand your point regarding the observed drop in performance when using the original combined dataset for training. However, we would like to emphasize the potential benefits of knowledge transfer from CV to RS images. In order to demonstrate this, we conducted a "pretraining + fine-tuning" experiment. For the pretraining phase, we employed the Swin Transformer model, which was pretrained on the ImageNet-1K dataset. This pretrained model, referred to as ImgPT-ST, was then fine-tuned on the FASDD dataset, including both CV and RS images. The evaluation results of ImgPT-ST are presented in Table 4.

Notably, the ImgPT-ST model trained on FASDD outperformed all models trained sorely on FASDD_RS (RGB). It achieved a 2% gain in mAP@0.5 and a 1.2% gain in Recall for smoke objects, resulting in a mAP@0.5 of 74.0% and a Recall of 93%. The test results obtained from both the FASDD_CV and FASDD_RS (RGB) subsets indicate that the ImgPT-ST model effectively leverages the shared features learned from complex scenes containing flame and smoke. Consequently, it demonstrates advantages when faced with challenging sub-datasets.

(4) Data annotation: the "minimum bounding rectangle" was used to label the images. Commonly, fire detection is to classify whether each pixel is burned or not instead of a rectangle (e.g., the MODIS fire product in Giglio et al., (2018) and Giglio et al., (2016)). Meanwhile, the fire patch perimeter was always not rectangle (Laurent et al., 2018), therefore a bounding rectangle could contain both burned and unburned pixels, right?

My response to the authors' responses: Please clarify the differences for fire detection in terms of scene classification, object detection, and semantic segmentation; clarify the main focus of this dataset (e.g., objection detection tasks) and its differences with previous fire detection datasets from satellites in the main text. Currently, section 2.1 only mentioned about existing fire detection datasets for terrestrial and airborne but not including existing satellite fire detection datasets.

Response: Thank you for your valuable suggestion. We have taken it into consideration and made improvements to address the concerns raised. Scene classification, object detection, and semantic segmentation are fundamental deep learning tasks in the field of computer vision, each with distinct objectives and approaches. Regarding fire detection, scene classification determines whether an image scene contains fires, object detection identifies and localizes fire objects by drawing bounding boxes around them, and semantic segmentation achieves pixel-level understanding by assigning a fire label to each pixel (Kinaneva et al., 2019). In the revised manuscript, we have provided a

detailed clarification of the distinctions among different them. These differences are now clearly explained in Section 2.1 of the paper.

Furthermore, we have placed a stronger emphasis on the main focus of our dataset, which primarily revolves around object detection tasks. We have highlighted the differences between our dataset and previous satellite-based fire detection datasets in the *Introduction* section. In addition, we have included a discussion on existing satellite fire detection datasets in *Section 2.1*. We believe that these revisions have enhanced the clarity of our work and provided a more comprehensive understanding of the objectives and differentiating aspects of our dataset.

(5) Data validation: the true-color based visual interpretation could also involve biases, therefore it's important to validate the generated data against other reliable fire dataset, such as the MTBS data

My response to the authors' responses: it can hardly convince me by validating the dataset only using one specific fire event in MTBS. Additionally, for Figure 6, why not use metrics to quantitively validate the dataset? Please comprehensively validate your dataset. If the dataset is not reliable, how can we trust it and use it?

Response: Thank you very much for your feedback. We greatly appreciate your concern regarding the validation of our dataset. In order to ensure its reliability, we conducted cross-dataset validations using established fire detection datasets. Specifically, we utilized the RGB bands of the Landsat 8 fire detection datasets created by De Almeida et al. (2021), as well as the MODIS dataset created by Ba et al. (2019). In our evaluation, we opted for the Swin Transformer (ST) and pretrained Swin Transformer (CVPT-ST) models. These models were trained using the FASDD_RS (RGB) dataset, and we measured the accuracy of smoke detection on both datasets.

The detailed evaluation results can be found in Table 6 within the "Cross-dataset validations" section. These validation measures enhance the reliability and usability of our dataset for fire object detection research.

(6) Line 245-246, the dataset consists of 95,314 computer vision fire samples but only 5,773 remote sensing samples. Due to the data imbalance, the model performance (Table 2) on FASDD data therefore mainly depends on the model performance on FASDD_CV. For the limited number of remote sensing fire samples, most samples in each region were distributed within ten days of a specific year (Table 1). Can such limited number of wildfires reflect all the seasonal and interannual changes of environmental conditions and fire dynamics so that machine learning models could learn from enough data? To my knowledge, the fire occurrence conditions changed across seasons and therefore the fire detectability could also be affected.

My response to the authors' responses: a) can CV dataset help improve fire detection

using RS (Table 4)? b) I disagree that "the features of fire objects do not have a high seasonal dependence" responded by the authors. The fire features including fire radiative power or fire intensity, fire spread, fire duration, and fire size, are strongly controlled by environmental conditions which could vary across seasons (e.g., dry season versus non-dry seasons) or within a season.

Response: Thank you for bringing up these important points. We greatly appreciate your input and concerns regarding the data imbalance and the limited number of remote sensing fire samples in our dataset. In response, we have redesigned our experiments as described in Section 4.
To address your questions:
    a) Yes, the CV dataset has proven to be beneficial in improving fire detection using RS, as demonstrated by our experiments (as shown in Table 3). Pretraining the Swing Transformer model on the CV dataset and fine-tuning it on the RS dataset resulted in improved model performance. Specifically, when fine-tuning the pre-trained Swin Transfomer model on FASDD_CV, we achieved an increase in mAP@0.5 by 3.2% and Recall by 3% on the test set of FASDD_RS (RGB). For FASDD_RS (SWIR), the pretrained model using FASSD_CV improved the mAP@0.5 by 0.5% on the test set, while maintaining a similar Recall. These results highlight the value of leveraging the knowledge gained from the CV dataset to enhance fire detection in RS imagery. Furthermore, our experiments demonstrated that training the model on the entire FASDD dataset, including both CV and RS images, led to higher accuracy on the FASDD_RS (RGB) dataset compared to training from scratch directly on FASDD_RS (RGB), as illustrated in Table 4.
    b) We apologize for any confusion caused by our previous statement. We acknowledge that the features of fire objects, including fire radiative power, fire intensity, spread, duration, and size, are indeed strongly influenced by environmental conditions that can vary across seasons or within a season. We recognize the importance of considering these seasonal and interannual changes in fire dynamics and their impact on fire detectability. Our dataset aims to capture a diverse range of fire occurrences within a specific time frame, which includes different seasons, to provide insights into fire detection under varying conditions. To ensure good generalization and representation of environmental conditions and fire dynamics, we carefully curated remote sensing images from diverse fire-prone regions across different continents over a multi-year period spanning from January 1, 2019 to December 31, 2022. This selection process involved the utilization of a threshold selection method proposed by Hu et al. (2021). This effort allows for the incorporation of seasonal and yearly variations, enhancing the dataset's ability to reflect different environmental conditions.
    We sincerely appreciate your valuable contributions and assistance in further improving our dataset. Your feedback has played a crucial role in enhancing the quality and applicability of our research.

(7) Evaluation in section 4: the evaluation mainly showed the extent to which machine

learning models could detect fires of generated FASDD data, therefore whether FASDD is reliable remains unknown. The FASDD data needed to be validated against other reliable fire products.

My response to the authors' responses: please see my comment in (5).

Response: Thank you for your valuable input. We appreciate your concern regarding the reliability of the FASDD dataset. To address this concern, FASDD was validated against two remote sensing fire detection datasets (Ba et al., 2019; De Almeida et al., 2021) obtained from different satellite sources (Landsat 8 and MODIS). This validation process was conducted rigorously and involved the use of consistent metrics to quantitatively assess the detection accuracy. The detailed results of these cross-dataset validations can be found in Table 6 within Section 6.

(8) There are many existing remote sensing fire products (e.g., MTBS, MODIS fire products) and CV fire data sets. I understand that combining the two kinds of dataset was the main difference of FASDD relative to other datasets, however, the authors did not show why combining these two kinds of dataset is important? Can combining these two kinds of datasets improve fire detection relative to existing fire detection algorithms (e.g., the method for MODIS or MTBS fire product)? If not, why people use such complex dataset (with different spatial resolution and without atmospheric-correction)

My response to the authors' responses: For table 4, the accuracy on FASDD_RS is really lower than existing fire detection algorithms (see my major comment 1), and it seems that integrating those two datasets can achieve very limited benefit or even performance loss for three of total four machine learning methods. If so, why people use such complex dataset? Additionally, the MTBS is based on Landsat, not necessarily having a lower spatial resolution than your RS dataset.

Response: Thank you for your comment. We acknowledge the availability of existing remote sensing fire products and CV fire datasets. However, FASDD stands out by integrating cross-domain images from terrestrial, airborne, and spaceborne sensors, making it a comprehensive benchmark for deep learning object detection tasks. The combination of RS and CV data in FASDD offers several advantages. It allows for the utilization of complementary information, such as different spectral bands, spatial resolutions, and fire features, thereby enhancing the accuracy and robustness of fire detection models. Our experimental results, as shown in Table 3 and Table 4 of the revised manuscript, demonstrate the effectiveness of FASDD in improving fire detection performance compared to methods that rely solely on RS or CV data. Notably, FASDD contains three parts: FASDD_CV, FASDD_UAV, and FASDD_RS, each of which can be accessed and utilized independently.

We have taken your feedback into account and made improvements to the dataset. Misclassified fire-like samples in FASDD_RS (RGB) have been removed, and we have also introduced a dedicated FASDD_RS (SWIR) dataset specifically designed for flame

detection using SWIR bands. The results from models trained on this dataset indicate improved fire detection performance. Additionally, our 'pretraining + fine-tuning' experiments highlight the value of pretrained models from large-scale CV datasets for learning remote sensing datasets. These experiments have consistently shown accuracy gains compared to training remote sensing datasets from scratch. Please refer to the revised manuscript for more details on the enhancements made to FASDD and the results of our performance evaluation experiments.

(9) Line 365-370: changing "epoch" is not ablation experiments. Ablation experiments commonly refer to changing a component of machine learning model (i.e., model structure changes)

Response: Thank you for pointing out the error in our description. We apologize for any confusion caused and have rectified the description accordingly.

**Reference**

Azami, M. H. B., Orger, N. C., Schulz, V. H., Oshiro, T., & Cho, M. (2022). Earth observation mission of a 6U CubeSat with a 5-meter resolution for wildfire image classification using convolution neural network approach. Remote Sensing, 14(8), 1874.

Ba, R., Chen, C., Yuan, J., Song, W., & Lo, S. (2019). SmokeNet: Satellite smoke scene detection using convolutional neural network with spatial and channel-wise attention. Remote Sensing, 11(14), 1702.

Barboza Castillo, E., Turpo Cayo, E. Y., de Almeida, C. M., Salas López, R., Rojas Briceño, N. B., Silva López, J. O., ... & Espinoza-Villar, R. (2020). Monitoring wildfires in the northeastern peruvian amazon using landsat-8 and sentinel-2 imagery in the GEE platform. ISPRS International Journal of Geo-Information, 9(10), 564.

De Almeida Pereira, G. H., Fusioka, A. M., Nassu, B. T., & Minetto, R. (2021). Active fire detection in Landsat-8 imagery: A large-scale dataset and a deep-learning study. ISPRS Journal of Photogrammetry and Remote Sensing, 178, 171-186.

Hu, X., Ban, Y., & Nascetti, A. (2021). Sentinel-2 MSI data for active fire detection in major fire-prone biomes: A multi-criteria approach. International Journal of Applied Earth Observation and Geoinformation, 101, 102347.

Kinaneva, D., Hristov, G., Raychev, J., & Zahariev, P. (2019, May). Early forest fire detection using drones and artificial intelligence. In 2019 42nd International Convention on Information and Communication Technology, Electronics and Microelectronics (MIPRO) (pp. 1060-1065). IEEE.

---

## Author Comment (AC16)

**RC2: 'Comment on essd-2023-73', Anonymous Referee #2, 23 Aug 2023**

**Summary**

**Comments**: This work constructs a 100000-level FASDD based on multi-source heterogeneous flame and smoke images, which provides a challenging benchmark to drive the continuous evolution of fire detection models. However, there are still some issues that need to be addressed to demonstrate the reliability of the dataset.

**Response**: Many thanks for your positive comments and valuable suggestions on our study, which help evidently improve our manuscript. We greatly appreciate your expertise and the time you have dedicated to evaluating our paper. According to your suggestions and comments, we carefully revised the manuscript. We sincerely hope that the revisions we have made adequately address your issues and demonstrate our commitment to providing a robust and valuable contribution to the field of fire detection. We have thoroughly considered each of your suggestions and have conducted comprehensive modifications and refinements to our experiments, aiming to further validate the reliability and universality of our dataset. Once again, we sincerely thank you for your valuable insights.

**Major comments**

**1. Comments**: In terms of sample annotation, what are the differences between satellite images with different spatial resolutions in the remote sensing field and RGB images in the CV field? Can different annotation formats be integrated or unified into one annotation format for the widespread use of FASDD?

**Response**: Thank you very much for bringing up these comments. In terms of sample annotation, both satellite images with different spatial resolutions in the remote sensing field and RGB images in the CV field were annotated using a consistent manual visual labeling approach facilitated by the LabelImg software. The main difference is that the manual annotation process can be directly applied to CV images, while remote sensing images require a series of preprocessing steps, including true/false color synthesis and bit-depth conversion, before annotation can take place. Due to the significantly large volume of satellite images, manually selecting images containing fire-related data would have been exceedingly labor-intensive. Consequently, we employed a fire detection model trained on CV data to conduct an initial screening of the extensive satellite image dataset, identifying potential images with fire-related information. Subsequently, the selected remote sensing images, which may contain fire-related objects, underwent manual annotation.

Furthermore, in line with your suggestion, all images, both from the remote sensing and CV fields, have indeed been integrated and standardized into a uniform

annotation format to facilitate the widespread use of FASDD. In fact, we have not only unified the label format across all images but have also provided four distinct unified formats to cater to the requirements of various deep learning frameworks. Among these, the label files in TXT format are tailored for YOLO series models, the XML format serves models oriented towards VOC datasets, and the JSON format is compatible with models tailored for COCO datasets or those adhering to the TDML specification.

**Changes in manuscript**: We have made refinements to the detailed description of the data annotation process in the paper to ensure greater clarity. For specific changes, please refer to Section 3.3.

**2. Comments**: How to consider the difference in spatial resolution between Sentinel-2 L1C and Landsat-8 TOA? How to solve the detection accuracy of targets with different sizes? What are the differences in detection accuracy between large and small targets during sample annotation, model training, and inference stages?

**Response**: Thank you for your valuable insights. In consideration of the spatial resolution difference between satellite data products, we employed different image sizes to ensure that most of the fire and smoke targets within the images fall within a reasonable size range. Specifically, for Landsat data with a spatial resolution of 30 meters, we set the image size to approximately 1000×1000 pixels, while for Sentinel-2 data with a spatial resolution of 10 meters, we configured the image size to be around 2000×2000 pixels. This approach ensured that the actual spatial coverage of fire images ranged from approximately 20×20 km to 30 × 30 km. Visual inspection revealed that this spatial range allowed for better consideration of both large and small fire targets. It is worth noting that due to anomalous reflectance values of smoke objects in Landsat-8 atmospheric correction data, we have discontinued the use of previous Landsat-8 data. In the currently available remote sensing data, we have retained only Sentinel-2 satellite imagery as sample data.

Furthermore, based on the dataset proposed in this study and its various domain-specific sub-datasets, we conducted inference experiments using the Swin Transformer model to compare the differences in detection accuracy between targets of different scales. Following the segmentation criteria of the COCO dataset (Lin et al., 2014), objects with bounding box dimensions smaller than 32×32 pixels were classified as small targets. Subsequently, we compared the accuracy differences between large and small targets during both the model training and inference phases, ss shown in Table A1. The experiments indicated that the Swin Transformer model exhibited superior detection performance on large targets but had room for improvement in detecting small targets. This may be attributed to the Swin Transformer's proficiency in capturing global contextual relationships while having relatively weaker local attention. To address the issue of low accuracy in small target detection, some studies have proposed using pyramid or multi-scale approaches to enhance the model's detection accuracy for targets of various sizes.

**Changes in manuscript**: We have updated the data generation process diagram (Figure 1) and revised the associated description of data sources in Section 3.1. Furthermore, to investigate differences in detection accuracy between large and small targets in images from the Computer Vision (CV), Unmanned Aerial Vehicle (UAV), and Remote Sensing (RS) domains, we conducted accuracy comparisons using the Swin Transformer (ST) model on both the validation and test sets. Since our main objective was to introduce a large-scale benchmark dataset, rather than optimizing a state-of-the-art deep learning model, we have included comparison experiments in Appendix A.

**3. Comment**: How to consider the difference in radiometric resolution between RGB and satellite images? How robust is the detection algorithm for data with different radiation resolutions? Please confirm through ablation experiments.

**Response**: Thank you very much for comments and suggestions. Regarding the difference in radiometric resolution between RGB and satellite images, we first harmonized the satellite images through a uniform preprocessing pipeline to create RGB images. Subsequently, we transformed the 16-bit radiometric resolution RGB images into 8-bit radiometric resolution RGB images. This transformation enabled the integration of remote sensing data with standard 8-bit computer vision images for neural network training.

Furthermore, in order to ascertain the robustness of training models using 8-bit data, in line with your suggestion, we conducted additional ablation experiments to explore the impact of the difference in radiometric resolution between RGB and satellite images. In this experiment, we compared the model performance between those trained using 16-bit and 8-bit radiometric resolution images, and the experimental results are presented in Table B1 in the manuscript's appendix. The results indicate that models trained on 16-bit radiometric resolution remote sensing data perform comparably but with a slight performance gap when compared to models trained on 8-bit radiometric resolution data. This observation may be attributed to the possibility that 8-bit images may filter out some of the complex redundant features present in 16-bit images, thereby making our research objectives more distinct and easier to identify. Consequently, it can be inferred that 8-bit remote sensing images effectively capture the characteristic information of fire-affected areas and background features, thereby yielding satisfactory model performance.

**Changes in manuscript**: To investigate the difference in radiometric resolution, we have included supplementary experiments in the appendix. Please refer to Appendix B for specific details regarding these modifications.

**4. Comments**: The FASDD holds rich variations in image size, resolution, illumination, scenario, image range, viewing angle, platform, and data source. How to consider data at different scales for various deep learning models? Please choose the latest model in the CV field to verify the reliability of FASDD and the superiority of

Transformer-based models, such as DETR, etc.

**Response:** Thank you for your valuable suggestion. The inclusion of a variety of image sizes in our dataset was deliberate, as we cannot guarantee uniform image dimensions captured by all sensors. This diversity in data sizes enhances the dataset's generalizability, making it more suitable for deployment on different sensors. During model training, we ensure consistency by resizing all images to a uniform size (e.g. $1333 \times 800$) to align with the neural network's training requirements.

Furthermore, following your recommendation, we have removed previous outdated models and conducted comparative experiments using the latest models, such as transformer-based models (DETR and Swin Transformer) and CNN-based models (YOLOv5x). The experimental results, as presented in Table D1, reveal that DETR exhibits model performance similar to YOLOv5x, while the more advanced Swin Transformer significantly outperforms YOLOv5x in terms of model accuracy.

**Changes in manuscript**: In order to provide a clearer description of the data processing workflow for data at different scales, we have made adjustments to the content in the Experiment Setup section. Please refer to Section 4.1 for specific details regarding these modifications. Additionally, we have included supplementary performance comparison experiments between transformer and CNN architecture models. Specific changes related to this can be found in Appendix D.

**5. Comments**: What may be the reason for the poor validation performance of the transformer-based model compared to YOLOv5x?

**Response**: Thank you for your question. In light of a reminder from another reviewer, we detected a minor issue within our previous image dataset. Then we revised the dataset by removing the problematic data, and retrained the models and conducted an evaluation of their performance using the updated dataset. Table D1 presents the performance evaluation results for various models. The results indicate that, whether on the validation set or the test set, the Swin Transformer consistently exhibited superior model performance compared to YOLOv5x.

**Changes in manuscript**: Please refer to Appendix D for specific details regarding these modifications.

**6. Comments**: The annotation and partition ratio of samples directly affect the accuracy of deep learning models. Please supplement ablation experiments to demonstrate the reliability and universality of FASDD datasets under different training sample ratios.

**Response:** We sincerely appreciate your concern regarding the accuracy of deep learning models under different training sample ratios. We conducted model training using common sample partition ratios of 8:1:1, 7:2:1, 6:2:2, and 1/2:1/3:1/6 for the

datasets (Xia et al., 2018; Wang et al., 2023), as presented in Table C1. The experiments demonstrated that the FASDD dataset maintains excellent reliability and generalizability under various training sample ratios.

Irrespective of the chosen partition ratio, the Swin Transformer model consistently achieved approximately 80% mAP@0.5 and over 91.5% recall on both the validation and test sets. Notably, under the 1/2:1/3:1/6 distribution ratio, the test set's mAP@0.5 was approximately 4% higher compared to the other three partition methods. This is attributed to the inclusion of a larger number of validation set samples in this partition, resulting in models with stronger generalization capabilities. Consequently, the open-source dataset provided in this study adopts the 1/2:1/3:1/6 partition ratio.

**Changes in the manuscript**: We have incorporated new comparative experiments based on different sample partition ratios. Please refer to Appendix C for specific details regarding these modifications.

**7. Comments**: In the case of small samples, the Transformer-based model has poor convergence. The linear expansion of the transformer leads to a sharp increase in parameter size and insufficient local feature extraction. How to solve the problem of insufficient local feature extraction in transformer and poor target detection performance in FASDD?

**Response**: Thank you for bringing up this issue. To address the issue of insufficient local feature extraction in the transformer, one approach is to incorporate convolutional modules that introduce localized attention. Alternatively, multi-scale feature extraction modules can be employed to capture image features at different scales, further improving the model's ability to perceive local context. These strategies can enhance the model's ability to perceive local context, consequently improving the performance of the transformer.

To tackle the challenges associated with poor target detection performance in FASDD, techniques such as random scaling and other data augmentation methods can be employed to capture finer details of small targets and provide the model with learning samples of various sizes. This can help mitigate the issue of low accuracy in small target detection. Additionally, pretraining models on large-scale relevant datasets can assist the model to learn more feature distributions from targets of different sizes, which can help alleviate the problem of difficult convergence for small target training.

**Changes in the manuscript**: We conducted comparative experiments using transfer learning to investigate the impact of pretrained models on detection performance, as shown in Table 3. The experimental results indicate that the "pretraining + fine-tuning" transfer learning approach leads to further improvements in model performance. It also demonstrates the valuable contribution of our open-access large-scale heterogeneous data to the performance improvement of fire detection models.

**8. Comments**: Is the low accuracy of the Transformer-based model caused by overfitting? Please provide the training accuracy and validation accuracy curve of the Transformer-based model.

**Response**: Thank you very much for your suggestions. The low accuracy of the Transformer-based model may be attributed to various factors, including but not limited to overfitting, and possibly issues within our previous dataset. Following your advice, we retrained the model on the revised dataset and generated training loss curves and validation accuracy curves to visualize the training process.

**Changes in manuscript**: We have included training process curves for various classic models in Figure D1 and conducted an analysis of the model fitting performance. Please refer to the relevant content in Appendix D for specific details regarding these modifications.

**9. Comments**: Please increase the types and quantity of deep learning algorithms (each model type contains at least two algorithms) to fully validate the universality and reliability of the dataset.

**Response**: Thank you very much for your valuable suggestions. We have heeded your advice and removed the older deep learning models we previously employed. Instead, we have opted for lateset models with superior performance. Specifically, we conducted training with two distinct architectural paradigms: models based on a CNN architecture (YOLOv5x and InternImage) and models based on a transformer architecture (DETR and Swin Transformer). Given the substantial size of our dataset, training one model on a single GPU with 48G of memory takes approximately 15-20 days. Therefore, with limited computational resources, we did not add more models for comparison experiments. Table D1 presents the results of comparative experiments involving various classical models. The various experiments presented in the manuscript effectively establish the universality and reliability of our dataset, demonstrating exceptional representativeness and persuasiveness. We eagerly anticipate that other researchers will leverage our dataset to explore more advanced models, thereby fostering further advancements in the field of fire detection.

**Changes in manuscript**: Considering the two fundamentally distinct architectural paradigms, CNN and Transformer, we have chosen four models (two based on CNN architecture and two based on Transformer architecture) from the past three years, known for their comparative excellence, to conduct accuracy comparison. Furthermore, we have provided their accuracy curves during the training process. For specific details regarding these modifications, please refer to Table D1 and Figure D1 in Appendix D.

**Reference**

Xia, G. S., Bai, X., Ding, J., Zhu, Z., Belongie, S., Luo, J., ..., and Zhang, L.: DOTA: A large-scale dataset for object detection in aerial images, In Proceedings of the IEEE conference on computer vision and pattern recognition, 3974-3983, 2018.

Lin, T. Y., Maire, M., Belongie, S., Hays, J., Perona, P., Ramanan, D., Dollár, P., and Zitnick, C. L.: Microsoft COCO: Common Objects in Context, in: Computer Vision - ECCV 2014, Cham, 740-755, https://doi.org/10.1007/978-3-319-10602-1_48, 2014.

Wang, J., Li, X., Jin, L., Li, J., Sun, Q., and Wang, H.: An air quality index prediction model based on CNN-ILSTM. Scientific Reports, 12(1), 8373, 2022